# DFT Study on Methanol Oxidation Reaction Catalyzed by PtmPdn Alloys

**Tingting Yang [1], Qian Xue [1,2], Xuewei Yu [1], Xueqiang Qi [1,3,\*], Rui Wu [2,\*], Shun Lu [4], Zhengrong Gu [4], Jinxia Jiang [5] and Yao Nie [6,\*]**

1   College of Chemistry and Chemical Engineering, Chongqing University of Technology, Chongqing 400054, China; Tingty@stu.cqut.edu.cn (T.Y.); xueq@stu.cqut.edu.cn (Q.X.); xueweiyu@stu.cqut.edu.cn (X.Y.)
2   Yangtze Delta Region Institute (Huzhou), University of Electronic Science and Technology of China, Huzhou 313001, China
3   School of Chemistry and Chemical Engineering, Chongqing University, Shazhengjie 174, Chongqing 400044, China
4   Department of Agricultural and Biosystems Engineering, South Dakota State University, Brookings, SD 57007, USA; shun.lu@sdstate.edu (S.L.); zhengrong.gu@sdstate.edu (Z.G.)
5   Chongqing Medical and Pharmaceutical College, Chongqing 400020, China; 2020364@cqmpc.edu.cn
6   Chongqing Key Laboratory of Green Synthesis and Applications, College of Chemistry, Chongqing Normal University, Chongqing 401331, China
\*   Correspondence: xqqi@cqut.edu.cn (X.Q.); ruiwu0904@uestc.edu.cn (R.W.); nieyao@cqnu.edu.cn (Y.N.)

**Abstract:** Pt is widely used as the catalyst for methanol oxidation reaction (MOR) in direct methanol fuel cells (DMFC). However, the high cost and limited supply of pure Pt limit the commercialization of DMFC. Herein, MOR catalyzed by variously designed Pd-doped $Pt_mPd_n$ was studied with the density functional theory (DFT); the $Pt_mPd_n(111)$ surface was chosen since it is the most stable surface among various low-index surfaces. The hydrogens in methyl groups were priorly dehydrogenated on Pt(111), followed by hydrogen in the hydroxyl group. The effects of both the ratio of Pt:Pd and the type of the alloy on the activity of $Pt_mPd_n$ catalysts toward MOR were also studied; both ordered and disordered PtPd with the 1:1 ratio had better catalytic activity towards MOR than other catalysts. Specifically, the disordered Pt:Pd$^d$ with the Pt:Pd ratio of 1:1 had the best activity for the relatively stronger adsorption of COH, but the lowest binding with CO and a moderate d band center. The adsorptions of both COH and CO are key steps in the MOR, since the steps of $CH_3OH \rightarrow CH_2OH \rightarrow CHOH \rightarrow COH$ have downhill energy profiles, while $COH \rightarrow CO$ is an uphill reaction. In addition, the d band centers of the surface atoms move towards the Fermi level with the increase of the Pd content; the d band can also be tuned by changing the atom arrangement. These findings can be used as rules to design high-performance catalysts for MOR.

**Keywords:** methanol oxidation reaction; density functional theory; PtPd alloys; catalysts; molecular modeling

## 1. Introduction

The direct methanol fuel cell (DMFC) has great potential in energy storage and conversion because it directly converts the chemical energy of methanol into electricity, with the merits of high-energy conversion efficiency and no pollutant emissions [1–5]. Pt is widely used as the catalyst for MOR due to its relatively high electrocatalytic activity [6]. However, the high cost and limited supply of Pt hamper the commercialization of DMFC. Furthermore, the strong adsorption of CO easily makes Pt poison, which will hinder the transfer of subsequent methanol molecules to the active site, resulting in sluggish reaction kinetics [7–11]. To decrease the Pt usage and improve the CO-poisoning tolerance, one appealing solution is the addition of other metal elements (such as Pd, Ru, Au, Ag, Mo, Co, Ni, Cu, and Sn) into PtM [12–19]. Among the pure metal catalysts, Pd is an effective

catalyst for MOR, with a good catalytic capacity [20]. Therefore, alloying Pt with Pd should be considered an effective approach to improve the activity of Pt-based catalysts for DMFC. The study on their synergistic effects as well as the mono-activities of pristine Pt or Pd atoms will be interesting.

At present, a large number of experimental and theoretical studies have confirmed the improved catalytic performances of PtPd bimetallic catalysts. For example, Higareda et al. [21] synthesized bimetallic PtPd nanoparticles and found that $PtPd_{0.5}$ with a Pt:Pd ratio of 2:1 has good improvement in CO poisoning resistance. Shen et al. [22] synthesized PtPd alloy nanowires supported on graphene and found that PtPd/graphene exhibited excellent electrochemical activity for MOR. Asmussen et al. [23] prepared nanostructured materials with different compositions of Pt, PtPd(25%), PtPd(50%), PtPd(75%), and Pd, finding that the PtPd(50%) catalyst had the highest electrocatalytic activity for the oxidation of methanol. The previous experiments confirmed that the alloyed PtPd catalyst had good catalytic activity and stability as an electrocatalyst for DMFC. However, the mechanism of methanol dehydrogenation on PtPd alloys is not clear. Xiao et al. [24] reported an alternate-reduction strategy, where the resulting PtPd hollow nanotubes had more favorable alloy impacts and a highly active (100) surface, confirming the excessive catalytic performance. Sakong et al. [25] studied the mechanism of $CH_3OH$ dehydrogenation on the Pt and distinguished characteristic differences between the methanol oxidation paths. Zhang et al. [26] studied the adsorption of intermediates involved in the methanol dehydrogenation process on the PdAu(100) surface. Xu et al. [27] investigated the oxidation mechanism of methanol ($CH_3OH$) on the surface of the PtPd(111) alloy, finding that alloying Pt with Pd could effectively improve the catalytic performance for $CH_3OH$ oxidation by changing the primary pathway from the CO pathway on pure Pt to the non-CO pathway on PtPd(111). You et al. [28] studied both PtAu(111) and PtPd(111), and the results showed that PtPd(111) has more active Pt sites for methanol oxidation and higher catalytic activity. A favorable pathway for methanol ($CH_3OH$) dehydrogenation on the $PtPd_3$(111) surface was identified, which followed the path as $CH_3OH \rightarrow CH_2OH \rightarrow CHOH \rightarrow CHO \rightarrow CO$, and the $PtPd_3$ catalyst with the Pt monolayer was verified as a good candidate material for methanol dissociation [29]. However, these works did not show (in-depth) the effects of both the ratio and the type of PtPd alloy. Therefore, the density functional theory (DFT) study was used to systematically investigate the mechanism of dehydrogenation of methanol on PtPd alloys with different types and ratios.

In this paper, the mechanism of methanol dehydrogenation on various PtPd alloys was investigated with DFT, and the optimal ratio of the PtPd alloy catalyst was screened. Among the low-index (100), (110), and (111) surfaces, the (111) surface with the largest coordination number was selected since it was the most stable. Based on the calculation of the adsorption of methanol as well as intermediates on the surface of Pt(111), it can be determined that methanol dehydrogenation tends to remove H in $CH_3$ first and then break O-H to form CO. Moreover, the ordered and chaos $Pt_mPd_n$ alloys with different ratios were studied comparatively. The corresponding adsorption energy, d band, and PDOS of PtPd alloys were explored. Specifically, the d band structure and charge transfer were analyzed in detail.

## 2. Computational Details

The DFT calculation was performed using the $DMol^3$ package integrated into the Materials Studio project [30–32]. The exchange–correlation energy was calculated using the Perdew–Burke–Ernzerhof (PBE) functional within the generalized gradient approximation (GGA) [33]. Meanwhile, a Fermi smearing method [34] with a smearing width of 0.01 Ha was used to accelerate the convergence. A double-numerical basis with polarization functions (DNP) was adopted to expand the valence electron functions. To take both computational accuracy and efficiency into account, the core treatment was set with density functional semicore pseudopotential (DSPP). In all the calculations, the vacuum was established at 15 Å to create a true crystal surface. The $3 \times 3 \times 1$ k-points were set

for all calculations, and the convergence threshold for the maximum force, maximum displacement, and energy were set to 0.004 Ha/Å, 0.005 Å, $2.0 \times 10^{-5}$ Ha, respectively. The $Pt_mPd_n(111)$ alloy was modeled by a four-layer slab with a p ($3 \times 3$) supercell. To explore the mechanism of methanol decomposition on various PtPd surfaces, methanol and intermediates were adsorbed on platinum and palladium in different proportions. During the calculations, the atoms in the two bottom layers were fixed in their bulk positions, while all other atoms were in completely relaxed states. The surface energy is calculated according to Equation (1),

$$E_{surf} = \frac{E_{slab} - NE_{Pt}^{bulk}}{2A} \tag{1}$$

where $E_{slab}$ is the calculated energy of the slab after relaxation of the structure, $N$ is the number of $Pt$ units in the slab, $E_{Pt}^{bulk}$ is the energy of a single $Pt$ unit in bulk $Pt$, and $A$ is the area of the surface supercell.

The adsorption energy ($E_{ads}$) of $CH_3OH$, $CH_3O$, $CH_2O$, $CHO$, $COH$, and $CO$ is defined as:

$$E_{ads} = E_{M/PtmPdn} - (E_{PtmPdn} + E_M) \tag{2}$$

where $E_{M/PtmPdn}$ is the total energy of the system with species adsorbed on $Pt_mPd_n$, $E_{PtmPdn}$ represents the energy of clean $Pt_mPd_n$, and $E_M$ means the energy of free species $M$. According to the above definition, a negative value indicates that the process is exothermic, whereas positive values indicate an endothermic process. A more negative $E_{ads}$ implies that the adsorption is thermodynamically more favorable.

The $d$ band center ($\varepsilon_d$) was calculated according to Equation (3),

$$\varepsilon_d = \frac{\int \rho E dE}{\int \rho dE} \tag{3}$$

where $E$ is the energy with respect to the Fermi level and $\rho$ is the electronic density of states.

## 3. Results and Discussion

### 3.1. The Determination of the Most Stable Surface of the Various $Pt_mPd_n$ Alloys

To screen out the most stable surface of the various $Pt_mPd_n$ alloys, the surface energy of eVery low-index surface was calculated. As shown in Table 1, all surface energies with (111) facets were no more than 0.10 eV, while they were between 0.10 and 0.12 eV for both (110) and (100) surfaces, indicating that the (111) surface is most stable. Thus, the $Pt_mPd_n(111)$ were chosen to be studied in this paper. In addition, it can be found that with the increase of Pd atoms in the alloy, the surface energy increases, which means that the exposure of Pt in various $Pt_mPd_n$ alloys would be more favorable. For the $Pt_mPd_n$ alloy with the Pt:Pd ratio of 1:1, both ordered and disordered (as depicted below in Table 1) were also studied. The surface energies for the same surface of the disordered PtPd were a little bit higher than that of the ordered ones. The electron density differences among Pt and Pd atoms were further studied, as shown in Figure 1, the interaction among the atoms on the (100) and (110) surfaces was much weaker than that on the (111) surface since the coordination number was less, leading to relatively larger surface energy. There is no significant accumulation or loss of electrons around both Pt and Pd for various $Pt_mPd_n$ alloys, indicating that Pt and Pd share comparable electronegativity.

**Table 1.** Surface energy of the low-index surfaces of various $Pt_mPd_n$ alloys.

| Systems | Surface Energy (eV/Å) | | |
|---|---|---|---|
| | **(111)** | **(110)** | **(100)** |
| Pt | 0.075 | 0.115 | 0.106 |
| $Pt_3Pd$ | 0.078 | 0.111 | 0.106 |
| $PtPd^d$ | 0.082 | 0.114 | 0.109 |
| $PtPd^{O1}$ | 0.078 | 0.113 | 0.107 |
| $PtPd^{O2}$ | 0.078 | 0.113 | 0.107 |
| $PtPd_3$ | 0.089 | 0.114 | 0.112 |
| Pd | 0.098 | 0.117 | 0.119 |

*Note:* $PtPd^d$ means the disordered PtPd alloy, $PtPd^{O1}$ means the ordered PtPd alloy, the outmost layer is Pt, $PtPd^{O2}$ means the ordered PtPd alloy, and the outmost layer is Pd.

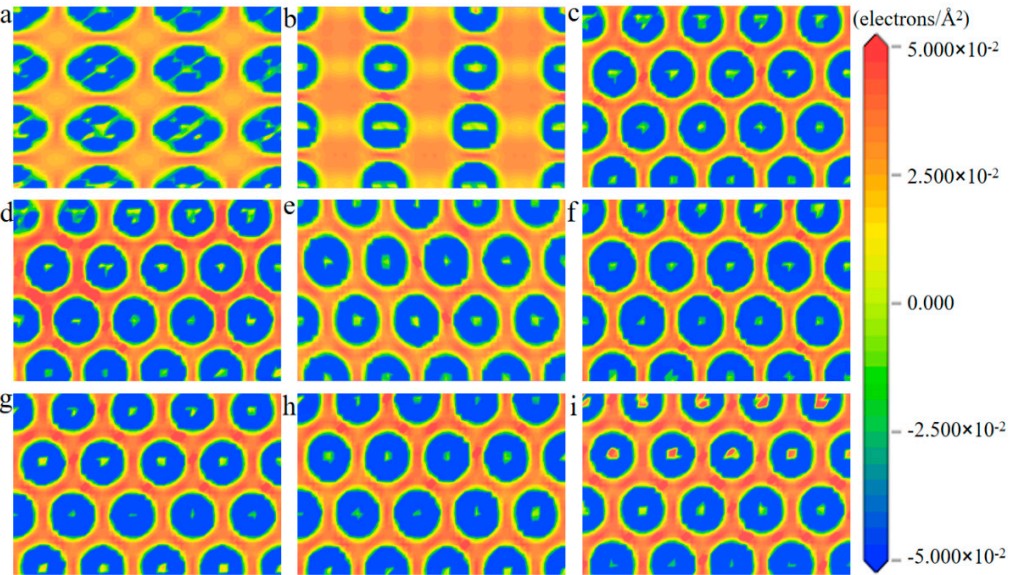

**Figure 1.** The slices (electrons/Å²) of the electronic density differences of (**a**) Pt(100), (**b**) Pt(110), (**c**) Pt(111), (**d**) $Pt_3Pd$(111), (**e**) $PtPd^d$(111), (**f**) $PtPd^{O1}$(111), (**g**) $PtPd^{O2}$(111), (**h**) $PtPb_3$(111), and (**i**) Pd(111) surface.

*3.2. The Properties of $Pt_mPd_n$ Alloy*

To examine the synergistic effects between Pt and Pd in the $Pt_mPd_n$ system, the d band centers of the $Pt_mPd_n$(111) surfaces were calculated. As shown in Figure 2, Pt has the lowest d band center with a value of −2.626 eV, while Pd has the highest d band center with a value of −2.040 eV. The d band centers of $Pt_mPd_n$ move closer to the Fermi level with the increase of Pd atoms. When the ratio of Pt:Pd is 1:1, the d band centers of $PtPd^d$, $PtPd^{O1}$, and $PtPd^{O2}$ are calculated to be −2.352, −2.532, and −2.238 eV, respectively. Specifically, the d band centers of $PtPd^d$ and $PtPd^{O1}$ are neither too high nor too low, and may possess better activity than the other alloys; the PtPd with the ration of 1:1 may be the most potential candidate as the MOR catalyst [35].

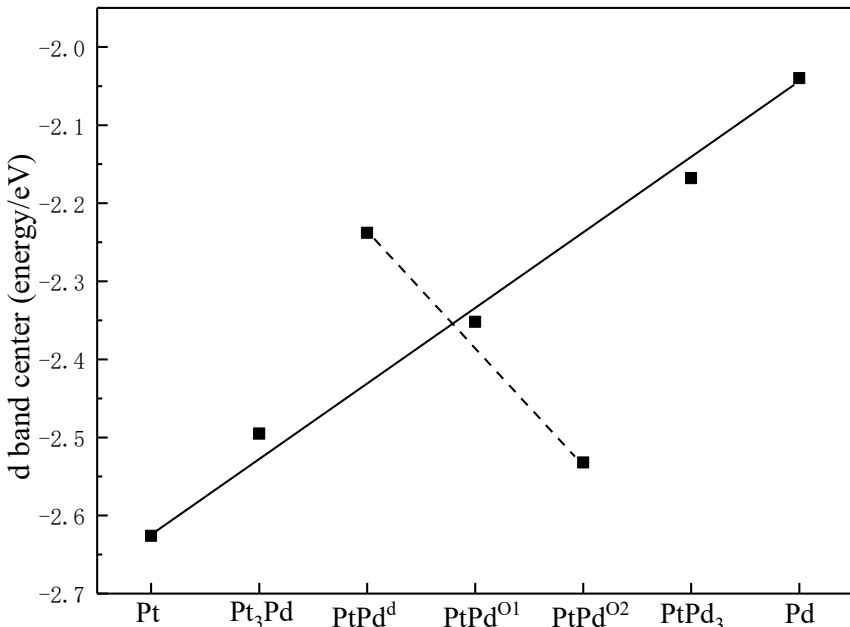

**Figure 2.** The d band center of atoms on the surfaces of Pt(111), Pt$_3$Pd(111), PtPd$^d$(111), PtPd$^{O1}$(111), PtPd$^{O2}$(111), PtPd$_3$(111), and Pd(111).

### 3.3. Methanol Oxidation Reaction on the Pristine Pt(111)

Methanol oxidation reaction (MOR) can proceed with either indirect and direct pathways on the Pt surface [36–38]. While CO will form after a series of dehydrogenation steps in the indirect pathway mechanism. Thus, all species adsorbed on the high-symmetry sites of Pt$_m$Pd$_n$(111) along the indirect pathway were studied here. As shown in Figures 3 and S1, four high-symmetry adsorption sites, including top (T), bridge (BRI), hexagonal close-packed (HCP), and face-centered cubic (FCC) sites on various (111) surfaces were considered.

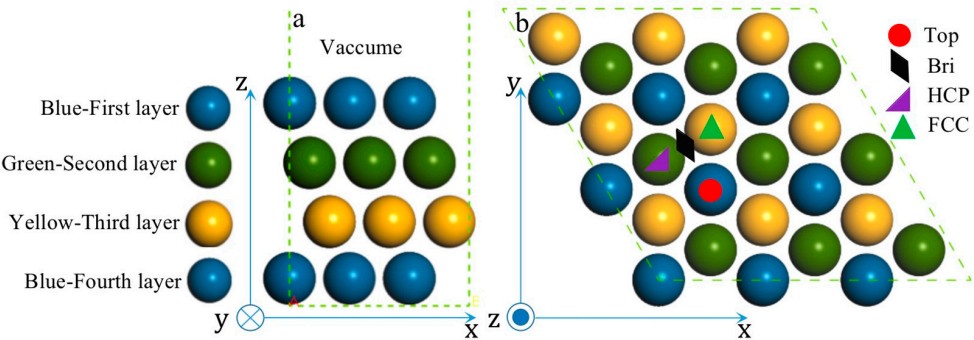

**Figure 3.** Side-view (**a**) and top-view(**b**) of pristine Pt(111).

The most stable configurations of the species in MOR adsorbed on pristine Pt(111) are presented in Figure 4; the detailed adsorption structures and energy are listed in Table 2 as well. For the adsorption of the methanol molecule, it can be found that methanol prefers to adsorb physically at the top site of Pt through the oxygen atom. The adsorption energy is only −0.404 eV with a distance of 2.470 Å between a surface Pt atom and the O in methanol. In addition, the bond lengths of C-H, C-O, and O-H in methanol are 1.099 Å, 1.440 Å, and 0.970 Å, respectively. The findings are in good agreement with the experimental values [39].

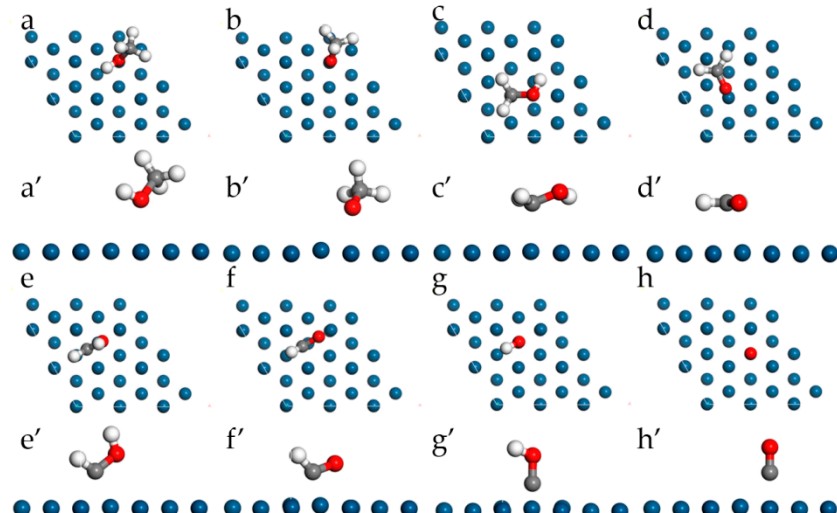

**Figure 4.** Top-view and side-view of the most stable configurations for MOR on pristine Pt(111). Methanol (**a**,**a′**), methoxy (**b**,**b′**), hydroxymethyl (**c**,**c′**), formaldehyde (**d**,**d′**), hydroxymethylene (**e**,**e′**), formyl (**f**,**f′**), hydroxymethylidyne (**g**,**g′**), CO (**h**,**h′**).

**Table 2.** The adsorption sites, adsorption energy ($E_{ads}$, eV), and bond lengths for the possible species of MOR on Pt(111).

| Species | Sites | Bond Length (Å) | Eads (eV) |
|---------|-------|-----------------|-----------|
| $CH_3OH$ | Top | d(C-O) = 1.440, d(O-Pt) = 2.470, d(C-H) = 1.099, d(O-H) = 0.970 | −0.404 |
| $CH_3O$ | Top | d(C-O) = 1.403, d(O-Pt) = 2.208, d(C-H) = 1.110 | −1.340 |
| $CH_2O$ | Top | d(C-O) = 1.216, d(C-Pt) = 2.575, d(O-Pt) = 2.261, d(C-H) = 1.119 | −0.165 |
| CHO | FCC | d(C-O) = 1.218, d(C-H) = 1.108, d(O-Pt) = 2.287 | −2.352 |
| $CH_2OH$ | Top | d(C-O) = 1.385, d(C-Pt) = 2.287, d(C-H) = 1.096, d(O-H) = 0.979 | −2.037 |
| CHOH | Bri | d(C-O) = 1.368, d(C-H) = 1.101, d(O-H) = 0.976 | −3.114 |
| COH | FCC | d(C-O) = 1.330, d(O-H) = 0.980 | −3.875 |
| CO | FCC | d(C-O) = 1.153, d(C-Pt) = 2.374 | −1.460 |

According to the MOR with an indirect pathway mechanism, four routes for methanol dehydrogenation to CO and H ultimately on the Pt(111) surface can be determined. Figure 5 lists the routes starting from both the initial O-H bond activation (route 1) and the initial C-H bond activation (route 2, 3, 4), separately. For the initial O-H bond activation in route 1, a series of $CH_xO$ intermediates could be determined. Methoxy ($CH_3O$) adsorbs stably at the top site with a chemical adsorption energy of −1.340 eV, and the Pt-O distance is 2.208 Å, which is 0.262 Å shorter than that in the adsorbed methanol molecule. However, the adsorption of $CH_2O$ is very weak and the adsorption energy is only −0.165 eV with the Pt-C and Pt-O distances are 2.575 and 2.261 Å, respectively, which means the dehydrogenation of $CH_3O$ to $CH_2O$ may thermodynamically be unfavorable. Formyl (CHO), and CO can be stably adsorbed at the FCC sites with the adsorption energies of −2.352 and −1.460 eV, respectively. For the initial C-H bond activation in routes 2, 3, and 4, a series of $CH_xOH$ intermediates have been calculated. In general, a decreasing C-O bond length can be observed with the decrease of H atoms in the methyl group. It has the smallest value when all the hydrogen is dehydrogenated, namely the CO molecule. Specifically, hydroxymethyl ($CH_2OH$) adsorbs at the top site with an adsorption energy of −2.037 eV and hydroxymethylene (CHOH) adsorbs at the bridge site with an adsorption energy of −3.113 eV. Considering that eVery species in route 4 is energy favorable, we propose that

the MOR starts with the break of the C-H bond in the methanol molecule and follows the CH$_3$OH→CH$_2$OH→CHOH→COH→CO pathway.

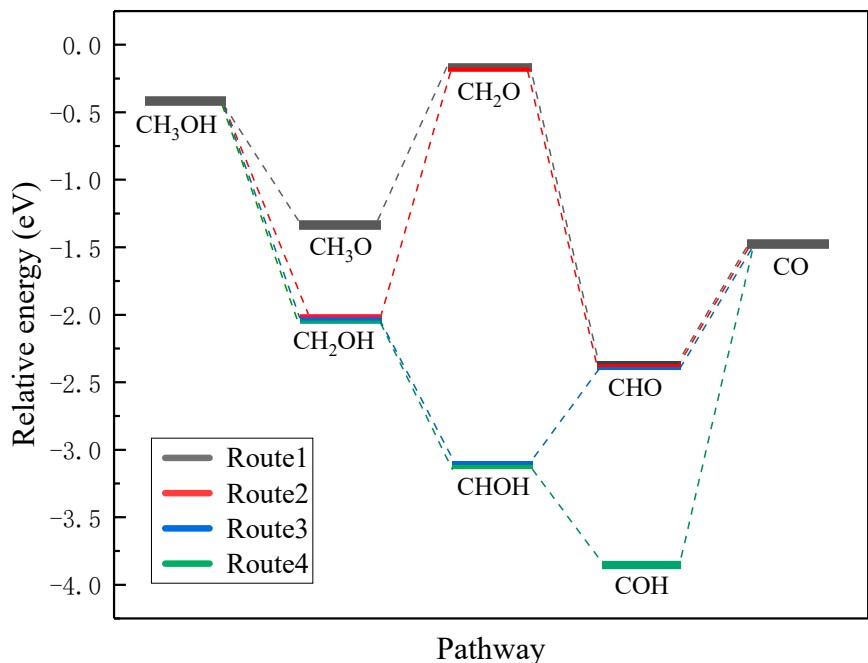

**Figure 5.** Four routes of methanol dehydrogenation on the pristine Pt(111) surface.

### 3.4. Methanol Oxidation Reaction on Various Pt$_m$Pd$_n$(111)

To compare the activities of various Pt$_m$Pd$_n$ catalysts, the thermodynamically favorable route 4 (CH$_3$OH→CH$_2$OH→CHOH→COH→CO) was studied further on various Pt$_m$Pd$_n$(111) surfaces. As illustrated in Figure 6 and Table S1, the methanol adsorptions do not show big differences from each other on eVery Pt$_m$Pd$_n$(111). While for the adsorption of CH$_2$OH and CHOH, Pt(111) has the strongest adsorption compared to the other surfaces, though the differences among the adsorption energy are still not very large. However, when the Pt:Pd ratio is 1:1, the adsorption of COH becomes very favorable, especially for the ordered PtPd$^{O2}$ alloy with the outmost layer of Pd. Furthermore, the disordered PtPd$^d$ alloy with a Pt:Pd ratio of 1:1 depicts the weakest adsorption for CO, and moderate adsorption for all intermediates. According to the Paul Sabatier principle [40], the random PtPd alloy with a 1:1 ratio may possess the best activity for MOR. In addition, as shown in Figure 7, the partial density of states (PDOS) of the methanol molecule and the intermediates adsorbed on the Pt, Pd, PtPd$^d$ alloys were comparatively studied, since the DOS can be effectively used to study the hybridization of the electron orbital between the adsorbed substance and the substrate [41]. Though the d orbitals of Pt, PtPd$^d$, and Pd show three sharp peaks and the methanol molecule has two sharp peaks ranging from −4.9 to −1.0 eV, the peaks are mismatched, indicating that only weak physical interactions exist between the adsorbed CH$_3$OH and the Pt(111) as well as the PtPd$^d$(111) and Pd(111) surfaces, consistent with the previously calculated adsorption energy. A similar phenomenon can also be observed for the adsorptions of CH$_2$O, which are also physical interactions. Being different from that of CH$_3$OH and CH$_2$O, the p orbital of the COH molecule and the d orbital of PtPd$^d$ overlaps significantly, ranging from −6.1 to −7.8 eV, which indicates a strong chemical interaction between the adsorbed COH molecule and Pt(111), as well as PtPd$^d$(111) or Pd(111) surfaces; the most overlap can be observed for the PtPd$^d$ catalyst. CH$_3$O, CH$_2$OH, CHOH, CHO, and the CO species more or less overlap, which means that the electron interaction between these intermediates and catalysts can be observed. For CO adsorption, it can be found that both the s and p orbitals of CO will interact with the d orbital of Pt$_m$Pd$_n$(111) around −7.8 and −5.8 eV since the matched peaks can be observed as shown in Figure 7h,p,x.

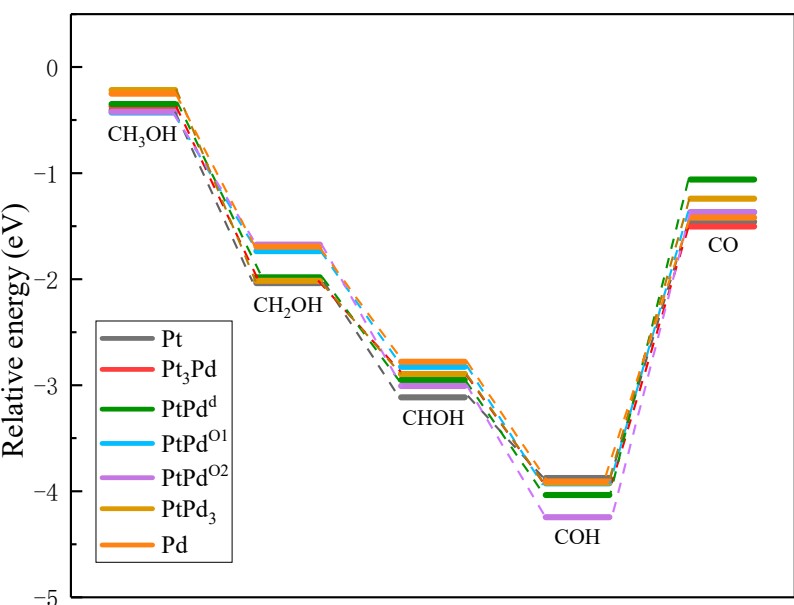

**Figure 6.** Adsorption energy of the optimal dehydrogenation path on Pt(111), Pt₃Pd(111), PtPd$^d$(111), PtPd$^{O1}$(111), PtPd$^{O2}$(111), PtPd₃(111), and Pd(111).

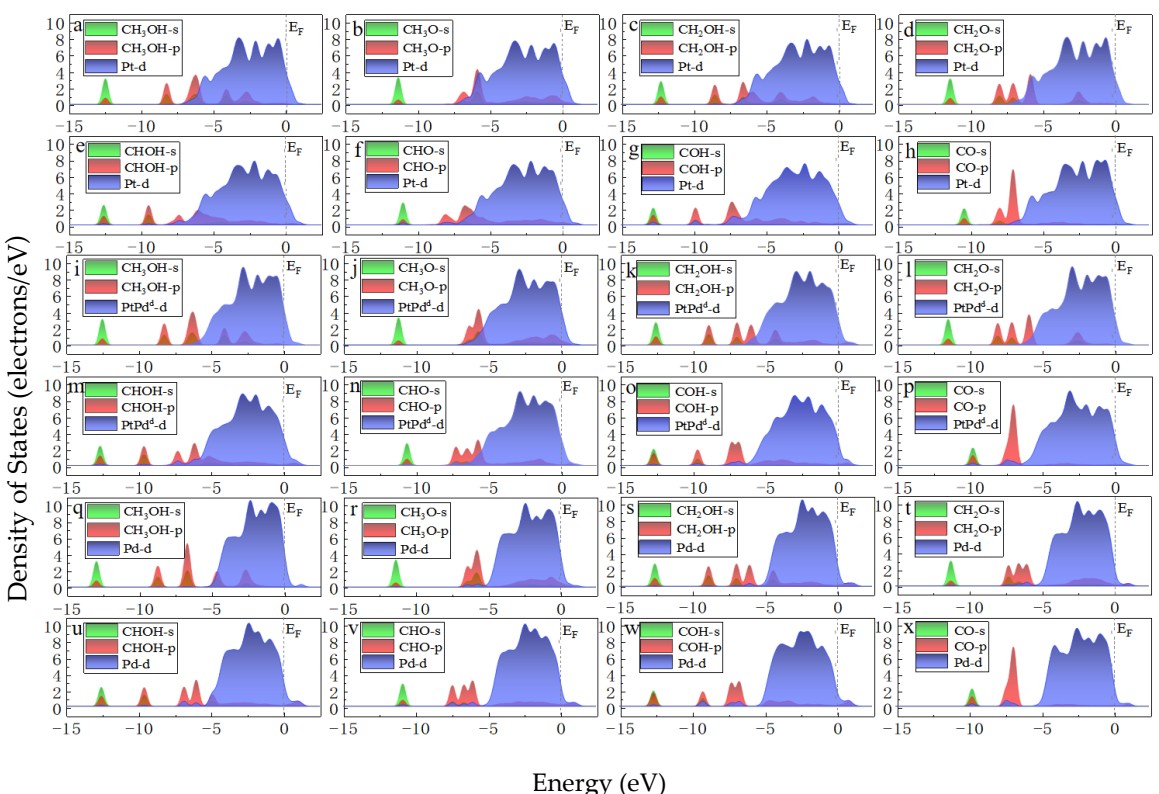

**Figure 7.** PDOS of CH₃OH, CH₃O, CH₂OH, CH₂O, CHOH, CHO, COH, CO adsorbed on Pt(111) (**a**–**h**), PtPd$^d$(111) (**i**–**p**), and Pd(111) (**q**–**x**). The Fermi level is set to 0 eV. The green and red represent the s and p orbitals of the methanol molecule, respectively. The blue represents the d orbital of the surface Pt$_x$Pd$_y$ (x + y = 4). (Pt$_x$Pd$_y$ means the closest four atoms around the adsorbed species on the first layer on pure Pt, PtPd$^d$, and pure Pd surfaces).

## 4. Conclusions

The MOR on various Pt$_m$Pd$_n$(111) surfaces with the indirect pathway mechanism was estimated by the periodic DFT calculation method. The Pt$_m$Pd$_n$(111) surfaces are most

stable since all the surface energies of $Pt_mPd_n(111)$ are no more than 0.10 eV, while they are more than 0.10 eV for both (110) and (100) surfaces. The possible methanol decomposition routes on the Pt(111) surface (by activating either O-H or C-H bonds) were further studied; it was found that the cleavage of the C-H bond of methyl takes precedence over that of O-H bond. The PDOS analysis and adsorption energy show that only $CH_2O$ adsorbed physically on the $Pt_mPd_n(111)$ surfaces, while other intermediates formed chemical adsorptions. The adsorption energy of $CH_2OH$ was more negative than that of $CH_3O$, which represents the main dissociation intermediate of methanol on the surface of Pt(111). Based on the compared results of adsorption energy, the optimal dehydrogenation path of methanol should be $CH_3OH \rightarrow CH_2OH \rightarrow CHOH \rightarrow COH \rightarrow CO$. The disordered $PtPd^d$ may be the best catalyst for MOR since it has the weakest adsorption for CO, moderate adsorption for the other intermediates, and a moderate d band center. The present study thus demonstrates that the PtPd alloy can effectively improve the catalytic efficiency of a single metal, but also reduce CO poisoning. This method can be used to understand and design other highly efficient catalysts for MOR.

**Supplementary Materials:** The following supporting information can be downloaded at: https://www.mdpi.com/article/10.3390/coatings12070918/s1, Figure S1: Various PtPd alloys: Pt3Pd (a), PtPdd(111) (b), PtPdO1(111) (c), PtPdO2(111) (d), and PtPd3 (e). The blue atom is Pt, and the green atom is Pd.; Table S1: Adsorption energy of the favorable route on various PtmPdn alloys.

**Author Contributions:** Conceptualization, X.Q. and Y.N.; methodology, T.Y., R.W. and X.Q.; software, T.Y. and Q.X.; validation, T.Y., Q.X. and Z.G.; formal analysis, T.Y., X.Y., J.J. and S.L.; investigation, T.Y. and Q.X.; resources, X.Q.; data curation, T.Y. and Q.X.; writing—original draft preparation, T.Y. and X.Q.; writing—review and editing, X.Q., R.W., Z.G., S.L. and Y.N.; visualization, Q.X.; supervision, X.Q. All authors have read and agreed to the published version of the manuscript.

**Funding:** This work was supported by the projects funded by the China Postdoctoral Science Foundation (2021M700621), 2021 Talent Introduction Project of Chongqing Medical and Pharmaceutical College (ygz2021104).

**Institutional Review Board Statement:** Not applicable.

**Informed Consent Statement:** Not applicable.

**Data Availability Statement:** Not applicable.

**Acknowledgments:** Z.G. thanks the support from NSF/EPSCoR (no. OIA-1849206, South Dakota 2D Materials for Biofilm Engineering, Science, and Technology Center (2DBEST)).

**Conflicts of Interest:** The authors declare no conflict of interest.

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
