# Peer review of "DFT Study on Methanol Oxidation Reaction Catalyzed by PtmPdn Alloys"

_coatings, doi:10.3390/coatings12070918_

Round 1
Reviewer 1 Report
This is clearly a case study of theoretical mathematical analysis with no motivation for a new physical problem and no new insight provided to a physical problem. In other words, the paper neither solves a physically relevant problem nor uses novel mathematical techniques which may have wider applications.
Author Response
Thank you for your comments, we tried our best to make the papers better. And all the modification have been highlighted in the revised manuscript. Please check the attachment.

Reviewer 2 Report
In the paper, Authors presented a quantum-chemical calculation about methanol oxidation reaction catalyzed by various designed PtmPdn alloys. Manuscript can be accepted for publication after a minor revision.
The manuscript is well written and prepared. Easy to read and follow. The following changes need to be made, in order to further improve the manuscript quality:
- Add “molecular modeling” to keywords.
- Would you explicitly specify the novelty of your work? What progress against the most recent state-of-the-art similar studies was made? Please refer to similar calculation work. Why is it important to understand the mechanism of this reaction? Please comment it and add information into manuscript.
- Please avoid lumping references (as in 38 line and all other). Instead summarize the main contribution of each referenced paper in a separate sentence.
- The English style should be revised.
Author Response
Thank you so much for your comments. All the modifications have been highlighted in the revised manuscript. Please check the attachment.

Reviewer 3 Report
There is one author in Suppl data that is not present in the main manuscript. Why is that so and please clarify.
Even though in the introduction is stated that conversion from CH3OH to CO is a downhill pathway, Figure 5 and 6 suggest otherwise. Please clarify.
In all 3D figures, a XYZ positioning reference must be improved
All figures have a very poor resolution and must be improved.
Figure 1 must have units of distance and quality must be improved.
Figure 4 has very low quality and seems like some atoms are cropped. Specifically, a to d seem to overlap.
The caption of figure 4 has a strange format with under and superscripts.
Figure 7 has 24 panes and very low resolution, font is minuscule and very hard to understand.
Since most of the discussion is base on the d-orbital influence, the shape and conjugation of these orbitals should be presented.
Author Response

(The authors gave the same response as above.)

Round 2
Reviewer 1 Report
accept for publication
Reviewer 3 Report
Figure 1 has been appropriately modified. However, there is low quality. Improve resolution
Figure 3. The figure is still recklessly cropped
Figure 7 is still long. Can you decrease y-axis scale?
Overall the text looks good and other issues have been corrected.